# The effects of a high-flavonoid corn cultivar on the gastrointestinal tract microbiota in chickens undergoing necrotic enteritis

Vinicius Buiatte[1], Monika Proszkowiec-Weglarz[2], Katarzyna Miska[2], Dorian Dominguez[3], Mahmoud Mahmoud[1], Tyler Lesko[4], Bryan P. Panek[4], Surinder Chopra[4], Mark Jenkins[5], Alberto Gino Lorenzoni[1] *

1 Department of Animal Science, College of Agricultural Sciences, The Pennsylvania State University, University Park, PA, United States of America, 2 Animal Biosciences & Biotechnology Laboratory, Beltsville Agricultural Research Center, USDA, ARS, Beltsville, MD, United States of America, 3 Veterinary Services, Animal and Plant Health Inspection Service, USDA, Richmond, VA, United States of America, 4 Department of Plant Science, College of Agricultural Sciences, The Pennsylvania State University, University Park, PA, United States of America, 5 Animal Parasitic Diseases Laboratory, Beltsville Agricultural Research Center, USDA, ARS, Beltsville, MD, United States of America

* agl20@psu.edu

**Data Availability Statement:** The obtained sequences were deposited in the NCBI Sequence

## Abstract

The search for alternative therapies to antimicrobial growth promoters (AGP) in poultry production has gained momentum in the past years because of consumer preference and government restrictions on the use of AGP in animal production. Flavonoids are plant-derived metabolites that have been studied for their health-promoting properties that could potentially be used as an alternative to AGP in poultry. In a previous study, we showed that the inclusion of a flavonoid-rich corn cultivar (PennHFD1) in the diet improved the health of broilers undergoing necrotic enteritis. However, the mechanisms of action by which the PennHFD1-based diet ameliorated necrotic enteritis are unknown. This study describes the microbial diversity and composition of the jejunum and ileum of chickens co-infected with *Eimeria maxima* and *Clostridium perfringens* and treated with a high-flavonoid corn-based diet. Luminal content and mucosal samples from the jejunum and ileum were collected for DNA extraction, 16S rRNA amplicon sequencing and data analyses. The infection model and the dietary treatments significantly changed the alfa diversity indices (Mucosal samples: ASVs, $P = 0.04$; Luminal content samples: ASVs, $P = 0.03$), and beta diversities (Mucosal samples: $P < 0.01$, Luminal content: $P < 0.01$) of the ileal samples but not those of the jejunal samples. The microbial composition revealed that birds fed the high-flavonoid corn diet had a lower relative abundance of *C. perfringens* compared to birds fed the commercial corn diet. The treatments also changed the relative abundance of other bacteria that are related to gut health, such as *Lactobacillus*. We concluded that both the infection model and the dietary high-flavonoid corn changed the broilers' gut microbial diversity and composition. In addition, the decrease in the relative abundance of *C. perfringens* corroborates with a decrease in mortality and intestinal lesions due to necrotic enteritis. Collecting different segments and sample types provided a broader understanding of the changes in the gut microbiota among treatments.

Read Archive (SRA) database (Accession number PRJNA955283).

**Funding:** This research was partially supported by a Hatch project (PEN04613) to SC and a Seed Grant Award from the College of Agricultural Sciences to GL and SC, and the in-house USDA-ARS CRIS project number 8042-31000-108-00D (MPW and KM).

**Competing interests:** The authors have declared that no competing interests exist.

## Introduction

Concerns with antibiotic resistance in animal production and market demands for poultry meat produced without antimicrobial growth promoters (AGP) have led a portion of the poultry industry to switch to antibiotic-free systems [1, 2]. Antibiotic stewardship in broiler production has greatly improved in the past years, with substantial reductions in the use of antibiotics [3]. However, the discontinued use of antibiotics is associated with an increased incidence of bacterial diseases in chickens, such as avian necrotic enteritis (NE) [1]

NE is an enteric disease caused by toxin-producing strains of *Clostridium perfringens*, a Gram-positive, spore-forming, and anaerobic bacterium ubiquitous in the gastrointestinal tract (GIT) of animals. NE affects chickens from 2 to 5 weeks of age, negatively impacting growth performance and increasing mortality [4]. It is estimated that NE costs $6 billion annually worldwide because of production losses and prevention measures [5]

Multiple factors predispose chickens to NE, such as *Eimeria* spp. infections, mycotoxins, dietary non-starch polysaccharides, and high dietary animal protein. These factors can result in increased mucus production, inflammatory responses, and an imbalance of the intestinal microbiota, favoring the multiplication and adhesion of pathogenic *C. perfringens* strains [6]

A healthy intestinal microbiota contributes to the host's physiology through several symbiotic mechanisms and outcompetes pathogenic bacteria that could disrupt homeostasis [7]. In chickens, this microbiota has fundamental roles that are important for growth performance, such as digestion and utilization of nutrients, modulation of the immune system, and protection against pathogens [8].

Alternative treatments that can support commensal bacteria and control pathogenic *C. perfringens* are needed to decrease the impact caused by NE. Phytobiotics are primary and secondary plant metabolites that have been studied as alternative candidates to antibiotics in poultry production [9]. Flavonoids are secondary plant metabolites that have been shown to have several health-promoting effects that support GIT health [10]. In humans, flavonoids have shown the potential to modulate the GIT microbiota, increasing the abundance of beneficial bacteria in the GIT [11]. In broiler chickens, flavonoid extracts can modulate the immune function and increase the abundance of beneficial microbes in the ceca [12].

In our previous study, the inclusion of a proprietary high-flavonoid corn cultivar (PennHFD1, Penn State University, USA) in the diets of broiler chickens coinfected with *E. maxima* and *C. perfringens* reduced the severity of necrotic enteritis. Chickens that received a diet with 31.5% of PennHFD1 had nearly 43% less mortality, 52% lower incidence of intestinal lesions, higher body weight gain, and lower feed conversion ratio (FCR) compared to chickens fed a diet formulated with a commercial corn variety [13]. However, the mechanisms of action by which the specialty corn ameliorated NE are unknown.

In this study, we hypothesized that the inclusion of PennHFD1 in the diets of the broiler chickens modulated the intestinal microbiota of chickens undergoing NE. The objectives of this study were to analyze the intestinal microbiota composition and diversity of chickens fed a PennHFD1-based diet and compare them to those of chickens fed a commercial corn-based diet.

## Materials and methods

### Experimental design

The experiment was conducted at the Poultry Education and Research Center, Penn State (University Park, PA), and has been previously published in Buiatte et al. (2022). All animal procedures were previously approved by the Institutional Animal Care and Use Committee

(**IACUC**) at The Pennsylvania State University (n. PROTO202001566). A total of 400 day-old straight-run broiler chickens (Ross 308, Aviagen) were randomly divided into 20 floor pens located in two identical temperature-controlled rooms to receive one of the following treatments: Uninfected birds fed a commercial corn-based diet (**CTL A**); Uninfected birds fed the PennHFD1-based diet (**CTL B**); Birds co-infected with *E. maxima* and *C. perfringens*, fed a commercial corn-based diet (**INF A**); Birds co-infected with *E. maxima* and *C. perfringens*, fed the PennHFD1-based diet (**INF B**). Treatments were assigned to floor pens using a completely randomized design with five replicates per treatment. Birds were reared for 21 days with feed and water provided *ad libitum* for the entire experiment. The treatment diets were formulated as previously described [13], to meet or exceed the National Research Council (1994). Corn represented 31.5% of the diet, and corn type (commercial corn vs. PennHFD1 corn) was the only difference between the diets. Birds were monitored twice a day throughout the experiment. The flavylium ion concentration was 1.98 absorbance/g for the PennHFD1 corn and 0.17 absorbance/g for the commercial corn cultivar. Proximate analyses of both corn lines were performed and nutrient composition was similar among the corn lines [13].

## Experimental model of necrotic enteritis

NE was induced by using a modified model previously described [14]. Briefly, birds were fed diets containing ingredients known to be predisposing factors for NE, such as wheat and fishmeal [15, 16]. At 13 days of age, chickens from the treatments INF A and INF B were infected with 5,000 oocysts of *E. maxima* via oral gavage. On days 18 and 19, the feed from the treatments INF A and INF B was inoculated with 1 mL of 1 x $10^9$ CFU of *C. perfringens* (2 NetB-positive strains, 1 NetB-negative strain) per bird. Feeders from all treatments were removed for 12 hours prior to the first inoculation of *C. perfringens*. On d 21, birds (5 birds/pen) were euthanized by cervical dislocation and necropsied for lesion identification and scoring (Data previously reported in Buiatte et al., 2022). All animals were euthanized at the end of the experiment by cervical dislocation.

## Sample collection

A subset of chickens was randomly selected for tissue sampling (n = ~5 birds/treatment). From each bird, four samples were collected for microbial composition and diversity analyses: jejunal luminal content (**JLC**); jejunal mucosa (**JM**), ileal luminal content (**ILC**), and ileal mucosa (**IM**). Jejunal samples were collected 10 cm proximal to the Meckel's diverticulum and ileal samples were collected 20 cm distal to the Meckel's diverticulum. For luminal content samples, the segment of the intestine was excised with a sterile scalpel blade, and content was collected directly into the cryogenic vials by applying manual pressure to the intestinal serosa. After the removal of intestinal content, the same intestinal segment was longitudinally opened to expose the mucosa. The remaining intestinal content was rinsed with sterile PBS, and the mucosa was scraped with a microscope slide (previously cleaned with 70% ethanol and DNA AWAY™, MBP, Inc., San Diego, CA) and stored in cryogenic tubes. All samples were immediately frozen in liquid nitrogen and transferred to a -80˚C freezer.

## 16S ribosomal RNA gene amplicon sequencing

DNA was extracted from the samples using the DNeasy PowerSoil kit (Qiagen, Valencia, CA, USA) according to manufacturer's instructions. DNA quantity was assessed with NanoDrop (ThermoFisher Scientific, Inc. Waltham, MA, USA), and DNA quality was evaluated with TapeStation System (Agilent Technologies, Santa Clara, CA, USA). High-throughput sequencing of the hypervariable region V3-V4 of the 16S ribosomal RNA gene was conducted with

Illumina workflow and consumables (Illumina, Inc., San Diego, CA, USA). The primers used for amplification of the target region were: Forward: 5′-TCGTCGGCAGCGTCAGATGTG TATAAGAGACAGCCTACGGGNGGCWGCAG-3′ and Reverse: 5′ GTCTCGTGGGCTCGGAGA TGTGTATAAGAGACAGGACTACHV GGGTATCTA ATCC-3′.

Amplification by PCR was followed by amplicon cleaning and indexing as previously described [17]. The concentration and quality of the amplicons were determined using QIAxcel DNA Hi-Resolution cartridge, proprietary QIAxcel ScreenGel software (version 1.6.0, www.qiagen.com), and QIAxcel Advanced System (Qiagen) following the manufacturer's instructions. The pooled DNA library (4nM) and PhiX control v3 (Illumina, Inc., 4nM) were denatured with 0.2 N NaOH (Sigma-Aldrich, Corp., St. Louis, MO, USA) and diluted to a final concentration of 4 pM. The library was mixed with PhiX control (20% v/v) and pair-ended 300 x 300 bp with Illumina MiSeq platform and MiSeq Reagent Kit v3 (Illumina, Inc). The obtained sequences were deposited in the NCBI Sequence Read Archive (SRA) database (Accession number PRJNA955283).

## Data analyses

Quality control and analysis of sequence reads were performed with Quantitative Insight Into Microbial Ecology (**QIIME**) software package 2 (version 2021.4.0, http://qiime2.org) [18]. Demultiplexing, filtering and dereplication of raw fastq files were performed with q2-dada2 [19]. Sequences with an average Phred score lower than 25 were removed from the dataset. MAFFT was used for multiple sequence alignment [20] and phylogenetic trees were generated with Fastree [21]. Taxonomy was assigned to amplicon sequence variants (**ASVs**) with DADA2 via the q2-feature-classifier classify-sklearn naïve Bayes taxonomy classifier [22] using the Greengenes database (v. 13_8; http://greengenes.secongenome.com).

Alpha diversity indices (ASVs, Shannon's diversity index, Pielou's Evenness, Faith's Phylogenetic Diversity) were obtained through QIIME2 package. Alpha diversity metrics were used to measure species richness and/or evenness within one sample, and the non-parametric Kruskal-Wallis test was used to analyze differences in alpha diversity between treatment groups. Analysis of beta diversity was performed in QIIME2 employing unweighted UniFrac. To test for significance in unweighted UniFrac distances, the non-parametric permutational analysis of variance (**PERMANOVA**) test was used. Principal coordinate analysis (**PCoA**) was used to visualize distances between treatment groups as well as to visualize clustering of samples (**QIIME2**) [23]. Linear Discriminant Analysis (**LDA**) Effect Size (**LEfSe**) algorithm [24] was used to identify taxa with significant differential abundance between treatment groups. Taxonomic composition and LEfSe graphs were generated using ggplot2 in R Studio (R Core Team, 2020).

## Results

### Sequencing data

A total of 11,655,786 raw sequences were generated from all samples (n = 88). After quality trimming, 3,138,150 sequence reads were obtained. The total number of raw and filtered reads and the mean reads per sample for each bacterial population are shown in Table 1.

Rarefaction curves were used to determine the sequencing depth for alpha and beta diversity analysis.

### Alpha diversity

The effects of treatments on alpha diversity indices are shown in S1 Table. Diet (Commercial corn or PennHFD1 diet) and infection (uninfected or coinfection with *E. maxima* and *C.*

**Table 1. Sequencing data from bacterial populations from luminal content (LC) and mucosa (M) of the jejunum (J) and ileum (I) in broiler chickens.**

|  | Sequence reads / sample type | | | |
|---|---|---|---|---|
|  | JLC[1] | JM[1] | ILC[2] | ILM[2] |
| **Raw reads** |  |  |  |  |
| Total | 1,344,406 | 1,936,943 | 4,036,054 | 4,338,383 |
| Mean | 64,019 | 92,235 | 175,480 | 188,625 |
| **Reads after quality trimming** |  |  |  |  |
| Total | 221,097 | 228,090 | 1,489,820 | 1,199,143 |
| Mean | 10,528 | 10,861 | 64,774 | 52,136 |
| Sequencing depth[3] for analysis | 2,600 | 1,699 | 12,133 | 2,701 |

[1] n = 21

[2] n = 23.

JLC (Jejunum luminal content); JM (Jejunal mucosa); ILC (Ileum luminal content); IM (Ileal mucosa).

*perfringens*) did not affect the alpha diversity in jejunal samples (JLC and JM, $P > 0.05$; S1 Table). In contrast, alpha diversity was significantly affected in ileal samples (ILC and ILM). The interaction of diet (commercial corn and PennHFD1 corn) and infection (uninfected and infected) significantly affected the ASVs of ILC samples ($P = 0.039$, S1 Table). Samples from infected birds fed a commercial corn diet (INF A) had fewer ASVs compared to CTL A and CTL B, but not different from INF B (Fig 1). In the IM samples, infected chickens fed the

**Fig 1. The effect of infection (control and infected) and diet (commercial corn and PennHFD1 corn) on amplicon sequencing variants (ASVs) of bacterial populations from ileal luminal content (ILC) in broiler chickens.** CTL A (Control birds fed the commercial corn diet); CTL B (Control birds fed the PennHFD1 diet); INF A (Birds co-infected with *E. maxima* and *C. perfringens* fed the commercial corn diet); INF B (Birds co-infected with *E. maxima* and *C. perfringens* fed the PennHFD1 diet).

commercial corn diet (INF A) had lower values of ASVs ($P$ = 0.045, Fig 2), less Richness ($P$ = 0.005, Fig 2), and lower Shannon index ($P$ = 0.038, Fig 2) in comparison to CTL A birds (Fig 2).

## Beta diversity

PERMANOVA analysis based on the unweighted UniFrac distances showed that treatments did not affect the beta diversity of microbial communities in jejunal samples (JLC and JM) ($P$ > 0.05, data not shown), but a trending separation between diets (commercial corn diet or PennHFD1 diet) was observed in JLC samples (PERMANOVA, $P$ = 0.055). The interaction between diet (commercial corn diet and PennHFD1 diet) and infection (control and infected) significantly affected the beta diversity of ileal luminal content (PERMANOVA, $P$ = 0.002) and ileal mucosa (PERMANOVA, $P$ = 0.003) samples (data not shown).

The principal coordinates analysis (PCoA) of microbial communities from JLC samples showed a trending clustering of treatments that received the PennHFD1 diet (CTL B and INF B) (Fig 3A). In JM samples, there were no apparent clusters among treatments (Fig 3B). The PCoA of bacterial communities in ILC samples did not show apparent clusters (Fig 3C). In IM samples, the CTL A samples clustered separately from the treatments CTL B, INF A, and INF B (Fig 3D).

## Microbial composition and differential relative abundance

The relative abundances of bacteria at the genus and species levels identified from jejunal and ileal samples are shown in Figs 4 and 5, and S2 Table.

**Jejunal samples.** *Lactobacillus* and *Clostridium* were the most relatively abundant genera in JLC samples (Fig 4A, S2 Table), from which *Clostridium* represented 18.76% of the genera identified in the treatment INF A and 3.32% of the genera identified in the treatment INF B. Regardless of the infection status, birds fed PennHFD1 showed a lower relative abundance of *C. perfringens* (CTL B, 0.79%; INF B, 3.26%) than chickens fed the commercial corn (CTL A, 2.40%; INF A, 18.76%; Fig 4B). From the *Lactobacillus* genera, *Lactobacillus salivarius*, and *Lactobacillus reuteri* were the most abundant species in JLC samples (Fig 4B). Lower abundance reads (LAR) and unclassified bacterial reads (UNCL) were less relatively abundant in infected treatments compared to uninfected treatments at the genus and species levels (Fig 4A and 4B).

More genera were identified in JM samples than in JLC samples, which included *Lactobacillus*, *Clostridium*, *Bacteroides*, and *Escherichia* (Fig 4C). Infected chickens obtained a higher relative abundance of *Escherichia* than uninfected chickens. LAR and UNCL were less relatively abundant in infected treatments than in uninfected treatments in JM samples (Fig 4C), but more relatively abundant in JM samples than JLC samples. At the species level, chickens fed PennHFD1 showed a lower relatve abundance of *C. perfringens* (CTL B = 0%, INF B = 0.40%; Fig 4D) than birds fed the commercial corn diet (CTL A = 1.95%; INF A = 12.27%; Fig 4D). *E. coli* and *L. reuteri* showed higher relative abundance in the infected birds than in the uninfected birds (Fig 4D).

**Ileal samples.** *Lactobacillus* was the most relatively abundant genera in ILC samples among all treatments (Fig 5A), and higher relative abundances were identified in chickens fed the commercial corn than in chickens fed PennHFD1. *Clostridium* and *Escherichia* were more relatively abundant in infected chickens compared to uninfected chickens, and at the species level, *C. perfringens* and *E. coli* were identified as the representative relatively abundant species (Fig 5B). Birds fed PennHFD1 showed a slightly lower relative abundance of *C. perfringens* than birds fed commercial corn.

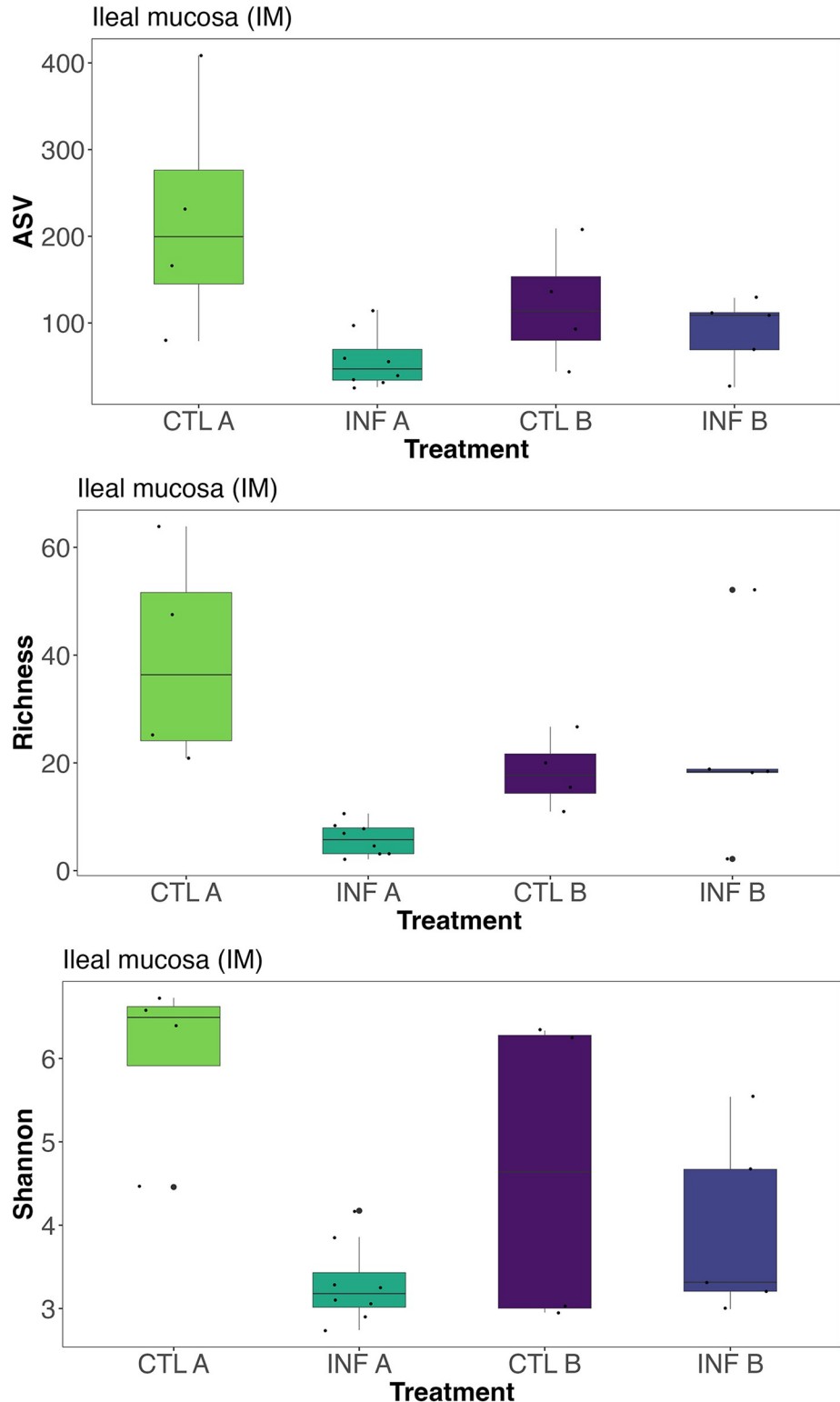

**Fig 2. The effect of infection (control and infected) and diet (commercial corn and PennHFD1 corn) on amplicon sequencing variants (ASVs), Richness and Shannon indices of bacterial populations from ileal mucosa (IM) in broiler chickens.** CTL A (Control birds fed the commercial corn diet); CTL B (Control birds fed the PennHFD1 diet); INF A (Birds co-infected with *E. maxima* and *C. perfringens* fed the commercial corn diet); INF B (Birds co-infected with *E. maxima* and *C. perfringens* fed the PennHFD1 diet).

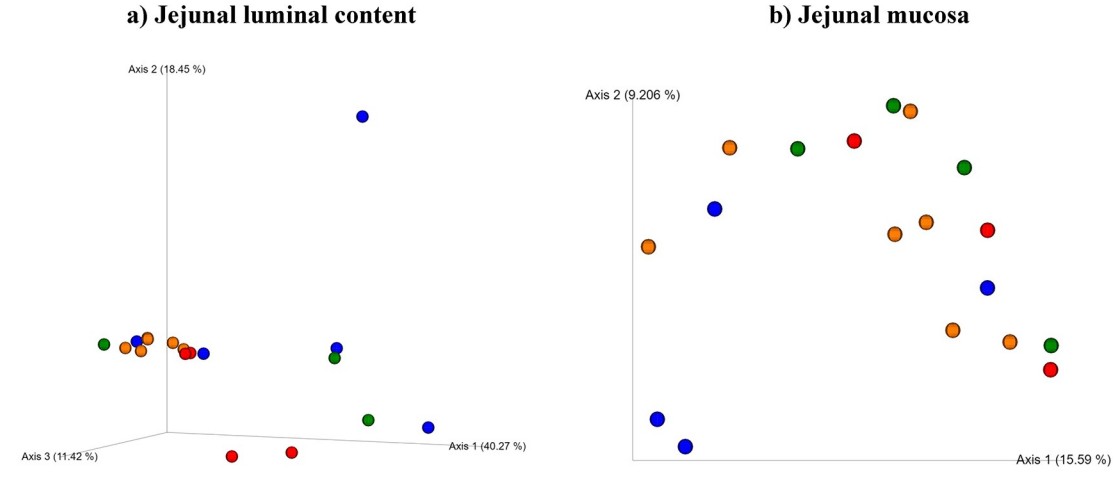

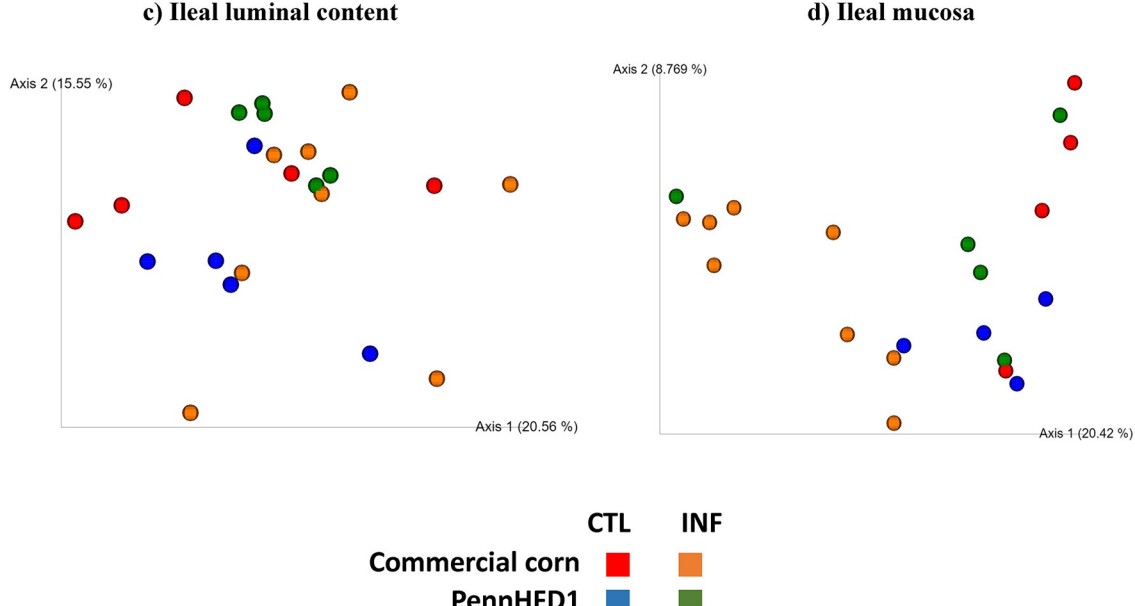

**Fig 3. Principal Coordinates Analysis (PCoA) based on unweighted UniFrac distances of the microbial communities found in ileal samples (ILC and IM) and jejunal samples (JLC and JM) collected from 21-day-old chickens coinfected with *E. maxima* and *C. perfringens*.** CTL (Control); INF (Infected); A (Commercial corn diet); B (PennHFD1 diet). Control: Non-infected chickens; Infected: Chickens coinfected with *E. maxima* and *C. perfringens*.

In IM samples, *Lactobacillus* was more relatively abundant in infected treatments than in uninfected treatments. At the genus level, LAR accounted for more than 40% of the reads obtained in the uninfected treatments (Fig 5C) compared to less than 10% in the infected treatments. *Clostridium* was less relatively abundant in chickens fed PennHFD1 (CTL B = 1.04%; INF B = 20.79%) than in chickens fed the commercial corn (CTL A = 2.55%; INF A = 26.72%) (Fig 5C). At the species level, *C. perfringens* was more relatively abundant in chickens fed the commercial corn (CTL A = 2.53%; INF A = 26.72%) than in chickens fed PennHFD1 (CTL

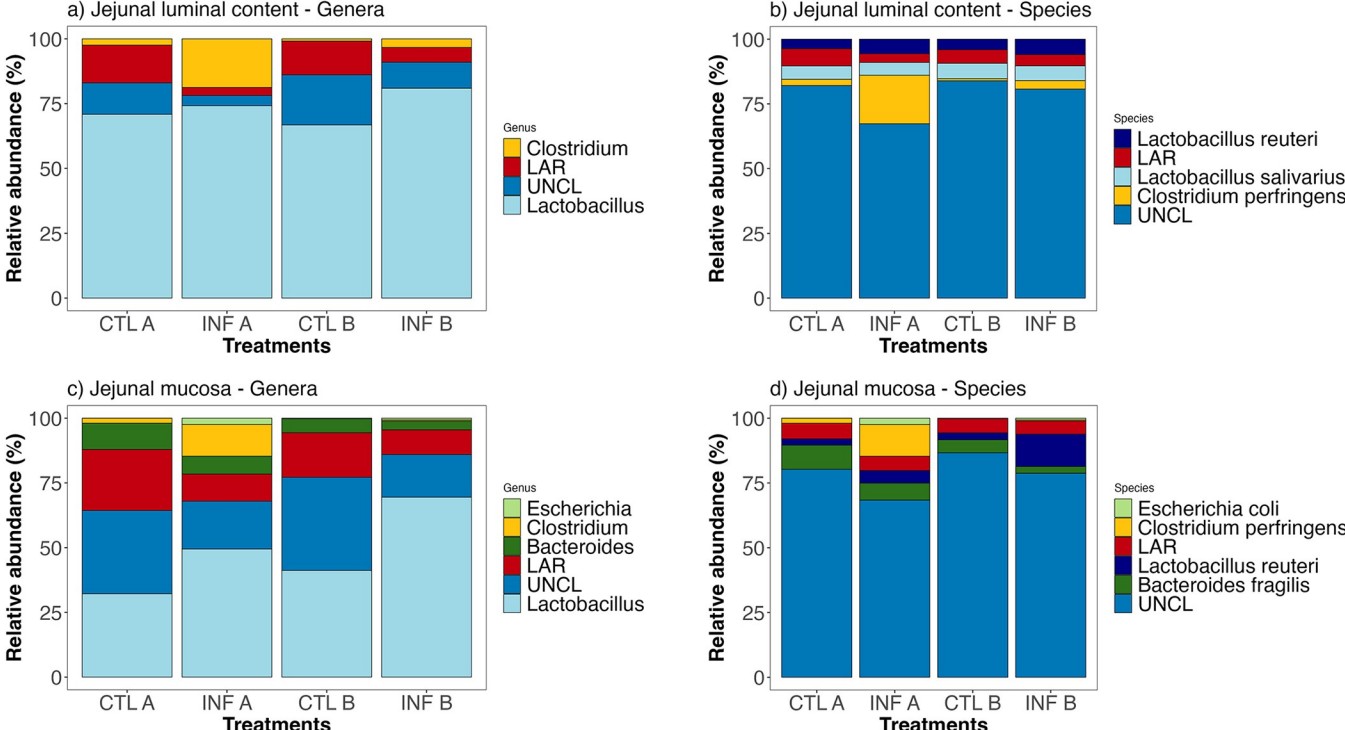

**Fig 4. Relatively abundant bacteria (%) identified in jejunal samples (JLC and JM) collected from chickens fed a commercial corn line-based diet or a high-flavonoid corn-based diet (PennHFD1), infected or uninfected (control) with *E. maxima* and *C. perfringens*.** JLC = jejunal luminal content; JM = Jejunal mucosa; Feed A = commercial corn line-based diet; Feed B = PennHFD1 (high-flavonoid)-based diet; Control (uninfected chickens); Infected (chickens co-infected with *E. maxima* and *C. perfringens*). UNCL = Unclassified bacteria reads; LAR = Low abundance reads.

B = 0.96%; INF B = 20.79%; Fig 5D). *Lactobacillus salivarius* and *Bacteroides fragilis* were more relatively abundant in the uninfected treatments than in the infected treatments (Fig 5D and S2 Table), whereas *Lactobacillus reuteri* was more relatively abundant in the infected than in the uninfected treatments.

**Differential relative abundance.** LDA effect size (LEfSe) identified the differentially abundant taxa when comparing the two diets (commercial corn and PennHFD1), regardless of the infection status. In JLC samples, differentially abundant taxa were identified in chickens fed PennHFD1, from which *Clostridium* was the most differentially relatively abundant genus (Fig 6). *Clostridium* was differentially abundant in JM samples from chickens fed the commercial corn diet compared to the PennHFD1 diet. At the family level, *Clostridiaceae*, *Enterobacteriaceae*, and *Oxalobacteraceae* were identified as differentially abundant. At the genus level, besides *Clostridium*, *Ralstonia* was also differentially abundant.

In ileal samples, differentially abundant taxa were identified in chickens fed PennHFD1 (Fig 7). In ILC samples, the family *Turicibacteraceae* was the most differentially abundant taxa, whereas, in IM samples, the genus *Streptococcus* was identified as the most differentially abundant bacteria.

## Discussion

Necrotic enteritis and dysbacteriosis are commonly associated because of a multitude of factors that can affect microbial balance in the intestines and the multiplication of *C. perfringens* [8]. Dysbacteriosis is commonly reported in chickens undergoing NE, and it has been shown to be an important consequence of the disease, and not necessarily a pre-existing condition

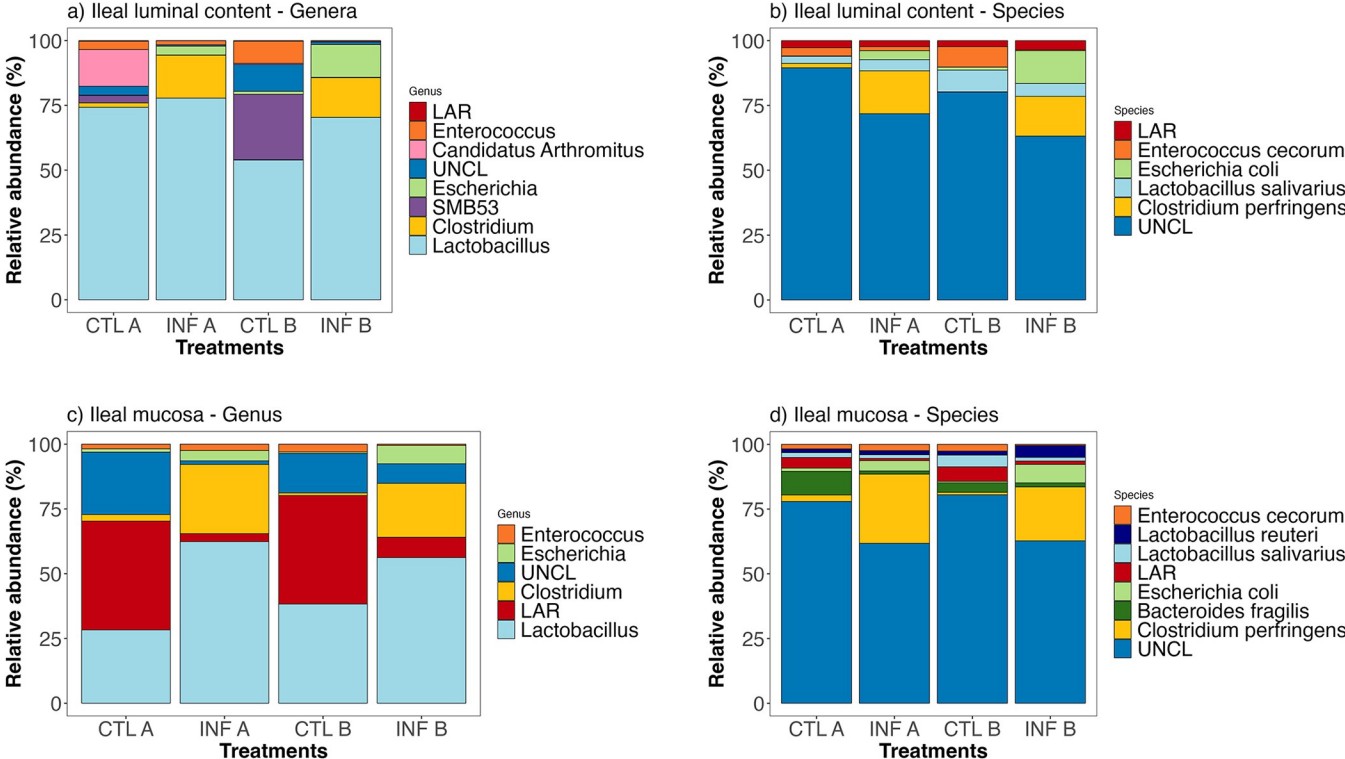

**Fig 5. Relatively abundant bacteria (%) identified in ileal samples (ILC and IM) collected from chickens fed a commercial corn line-based diet or a high-flavonoid corn-based diet (PennHFD1), infected or uninfected (control) with *E. maxima* and *C. perfringens*.** ILC = Ileal luminal content; IM = Ileal mucosa; Feed A = commercial corn line-based diet; Feed B = PennHFD1 (high-flavonoid)-based diet; Control (uninfected chickens); Infected (chickens co-infected with *E. maxima* and *C. perfringens*). UNCL = Unclassified bacteria reads; LAR = Low abundance reads.

[25]. Displacement of some bacteria, such as lactic acid-producing bacteria and butyrate producers is commonly reported in NE studies that use predisposing factors alongside inoculation of *C. perfringens* [26].

Many studies have shown that NE challenges can differentially impact the microbiota. For instance, some NE models reduce the abundance of *Lactobacillus* in the small intestines, which can be interpreted as an important finding of the disease because of the role of *Lactobacillus* in GIT health and function [27, 28]. In our experiment, birds co-infected with *E. maxima* and *C. perfringens* showed a slightly higher relative abundance of *Lactobacillus* compared to uninfected chickens. Bortoluzzi et al. (2019) [29] reported an increase in *Lactobacillus* abundance in the ileum of birds challenged with a model of NE that used a coccidia vaccine and reused litter. A study that characterized the cecal microbiota of chickens under four NE models, obtained an increased relative abundance of some *Lactobacilli* species in birds that received *Eimeria* as part of the NE model, which was correlated with an increased concentration of short-chain fatty acids (SCFAs) in the GIT [30]. These differential results highlight the variability of the GIT microbiota in response to different NE challenges.

Polyphenolic compounds, such as flavonoids, can be metabolized by intestinal bacteria. Studies have shown that some polyphenols can modulate the GIT microbiome by promoting the multiplication of beneficial bacteria and suppressing pathogens [31]. In our previous experiment, a flavonoid-rich corn (PennHFD1) diet ameliorated the impacts caused by NE in broiler chickens [13]. In this study, the effects of the PennHFD1-based diet on the intestinal microbiota of those chickens were evaluated.

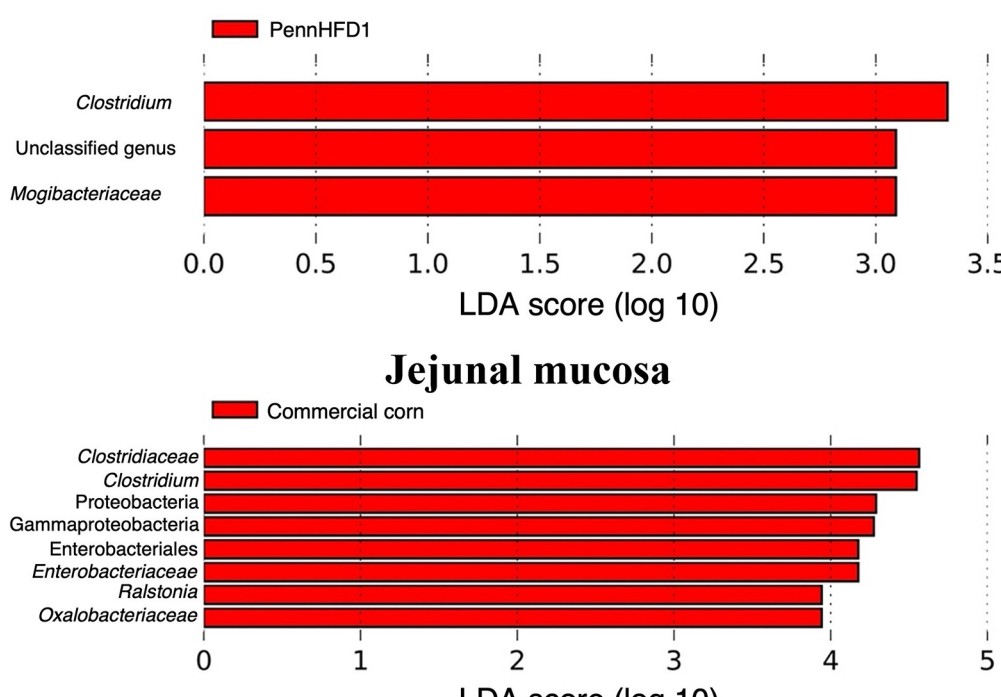

**Fig 6.** Effect of diet (A and B) on differentially abundant taxa in jejunal samples (JLC and JM) obtained by Linear Discriminant Analysis (LDA) effect size (LEfSe). Commercial corn = Commercial corn-based diet; PennHFD1 = PennHFD1-based diet; ILC = Ileal luminal content samples; IM = Ileal mucosa samples.

The treatments (infection and diet) did not change the alfa diversity of jejunal samples but significantly changed those of ileal samples, possibly because ileal samples had a richer microbial composition. For instance, in JLC samples, the most abundant genera were *Lactobacillus* and *Clostridium*, whereas in ILC samples, the most relatively abundant genera were *Lactobacillus*, *Clostridium*, *Enterococcus*, *Escherichia*, *Candidatus Arthomitus* and SMB53. Therefore, a greater number of genera could have been susceptible to effects caused by the infection model that involves *E. maxima* and *C.* perfringens, and the dietary components, leading to changes in the alfa diversity indices.

The corn type and infection status had a confounding effect on the beta diversity of ileal samples. Dietary components and pathogens are among the most important factors that can drive differences in the GIT microbiota [31]. A trending difference in the unweighted UniFrac distances between the diets (PennHFD1 and commercial corn) was observed in JLC samples using PERMANOVA and PCoA, which could have been significant with a larger sample size.

Based on the taxonomic classification, *Lactobacillus* was the most relatively abundant genus in both jejunal and ileal samples. Luminal samples (JLC and ILC) had a higher relative abundance of *Lactobacillus* than mucosal samples (JM and IM). In fact, *Lactobacillus* is among the most abundant bacteria found in the small intestines of chickens [32, 33]. *Lactobacillus* is commonly associated with healthy microbiota, and it is included in several probiotic products designed to improve growth performance in chickens [34]. In their symbiotic relationship with the host, *Lactobacilli* produce SCFA and lactic acid from nutrients released during

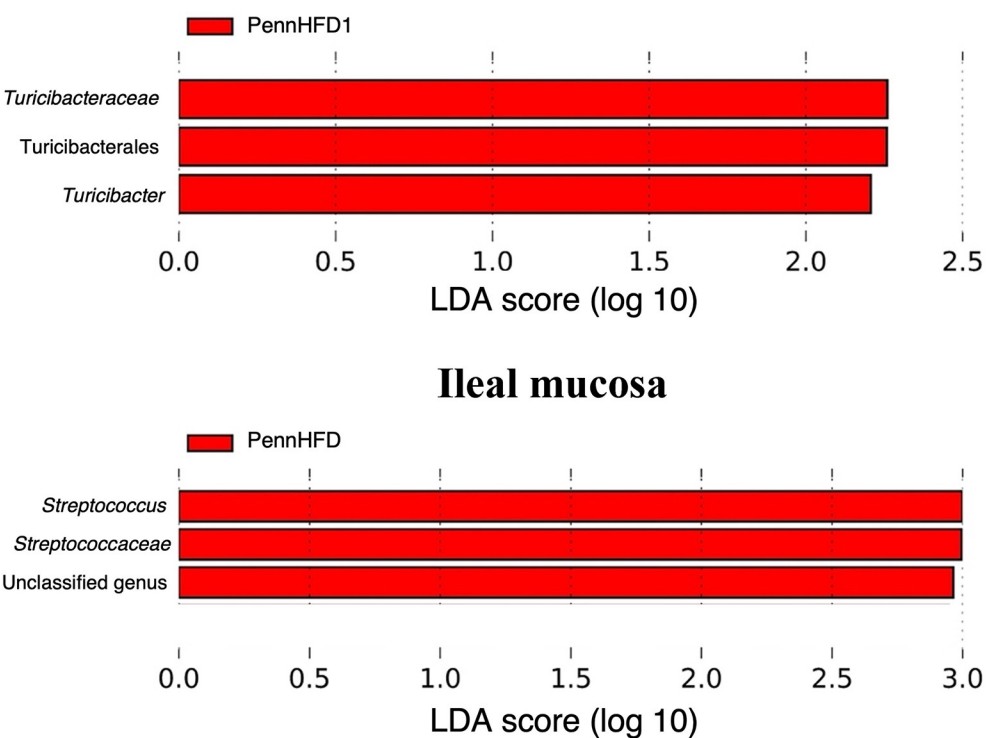

**Fig 7.** Effect of diet (A and B) on the differentially abundant taxa in ileum samples (ILC and IM) obtained by Linear Discriminant Analysis (LDA) effect size (LEfSe). Commercial corn = Commercial corn-based diet; PennHFD1 = PennHFD1-based diet; ILC = Ileal luminal content samples; IM = Ileal mucosa samples.

digestion. The production of these metabolites leads to a decrease in the luminal pH, which boosts the multiplication of *Lactobacilli* [34–36] and may inhibit pathogenic bacteria, such as *C. perfringens*. Previous studies show the antagonistic effect of *Lactobacillus* against *C. perfringens* through the production of bacteriocins and competitive colonization of the intestines [37]. As previously mentioned, the relative abundance of *Lactobacillus* was higher in infected chickens than in uninfected chickens, and no consistent effect of the diets on *Lactobacillus* was observed.

In all sample types, *C. perfringens* was less relatively abundant in chickens fed the high-flavonoid corn diet than in chickens fed the commercial corn diet, regardless of infection status. In a murine study, animals fed a diet with a flavonoid-rich ingredient (*Rubus occidentalis*) had less *Clostridium* spp. in the colon luminal content compared to those fed the control diet [38]. Some classes of flavonoids, such as flavan-3-ols, have shown antimicrobial activity against *C. perfringens in vitro* [39]. Interestingly, the genus *Clostridium* was identified as more differentially relatively abundant (LEfSe; Fig 6) in the luminal samples of the jejunum (JLC) from chickens fed the PennHFD1 diet compared to chickens fed the commercial corn, regardless of the infection status. Conversely, in mucosal samples (JM), *Clostridium* was identified as more differentially abundant in chickens fed the commercial corn diet (LEfSe, Fig 6). The formation of NE lesions occurs when *C. perfringens* attaches to the enterocytes and secretes proteolytic toxins [40]. It has been shown that after the formation of lesions, *C. perfringens* lines up at the submucosa forming a biofilm-like structure [4]. Therefore, mucosal samples may be more

appropriate to evaluate *C. perfringens* infection than luminal content samples with high-throughput sequencing techniques.

Although *Clostridium* comprises many species that are not related to necrotic enteritis, reports show that inoculation of chickens with pathogenic strains of *C. perfringens* leads to a dramatic displacement of the native *Clostridia* in the intestine [25]. In our experiment, chickens were infected with a mixture of toxin-producing strains, isolated from field cases of NE. Although *C. perfringens* was also identified as relatively abundant in uninfected treatments, no intestinal lesions were detected in the sampled birds [13]. Therefore, in the infected treatments, *C. perfringens* may have more effectively colonized the intestinal mucosa of chickens fed commercial corn than of chickens fed PennHFD1. This is associated with the incidence of intestinal lesions; chickens fed the PennHFD1 diet had a 52% lower incidence of intestinal lesions compared to chickens fed the commercial corn diet [13].

It should be noted that PennHFD1 may have interfered with several factors intrinsic to our experimental model, that could have influenced the gut microbiota, such as *E. maxima* infection and replication, the intestinal immune response, as well as other physiological processes. It has been shown that supplementation of flavonoids in broiler diets can modulate cellular and humoral immune responses, and lipid metabolism, and promote anti-oxidant effects [41]. Moritz et al. (2022) [42] showed that birds fed a sorghum-based diet containing several flavonoid compounds were less impacted by NE, which was associated with the upregulation of genes involved in the immune response to bacterial infections; however, the authors did not evaluate possible effects against *E. maxima*. In a previous study of our research group involving a different experimental design, we observed that the inclusion of PennHFD1 in a corn-soy based diet did not interfere with *E. maxima* oocyst shedding [43]. The complexity of the host-microbiota crosstalk and the onset of NE should be considered when interpreting the beneficial effects of feed additives and functional feedstuffs in experimental infections.

Among other taxa reported in this study, *Candidatus Arthromitus* was displaced in the ileal samples of birds co-infected with *E. maxima* and *C. perfringens*. These are segmented filamentous (SFB) commensal to the intestinal mucosa. They play a key role in the innate immune system, inducing Th17 lymphocytes to produce IL-17, a proinflammatory cytokine that increases during *E. maxima* infections [44]. In our study, *C. Arthromitus* was only identified in uninfected chickens. A similar finding was observed in a previous study that used three broiler breeds infected with *E. maxima* and *C. perfringens*, in which *C. Arthromitus* was only identified in uninfected Ross chickens. The authors suggested that this could be related to a possible difference in the resistance to NE among breeds [45]. However, data from other studies suggest that factors that interfere with the GIT lining integrity, such as mycotoxins and *Eimeria* infections, may negatively affect *C. Arthromitus* which could interfere with immunity development [26].

*Escherichia* is among the most abundant genera found in the intestines of chickens. Studies using polyphenolic compounds have found various effects on *Escherichia* [31]. In our study, *Escherichia* was more abundant in infected chickens than in uninfected chickens. In ileal samples, chickens fed the high-flavonoid corn showed a higher abundance of *Escherichia* than chickens fed the commercial corn. High *Escherichia* counts have been correlated with a low FCR in chickens [46]. In the current study, infected chickens fed high-flavonoid corn had lower FCR compared to infected birds fed a commercial corn diet [13].

The diversity and composition of the GIT microbiota change substantially throughout different segments of the chicken GIT [47]. Moreover, studies have shown remarkable differences between the microbiota of luminal content samples and mucosal samples [48, 49]. In jejunal samples, *Escherichia* and *Bacteroides* were only identified as relatively abundant in mucosal samples and not in luminal content samples, and several taxa were differentially abundant in JM samples compared to JLC samples (Fig 6). In the ileum, *Candidatus Arthromitus* was only

identified in the luminal content samples and not in the mucosal samples. In ILC samples, *Turicibacter* was significantly influenced by the diet, whereas in IM samples, the dietary treatment significantly affected *Streptococcus*. The collection of different intestinal segments and sample types provided a broader picture of the effect of the treatments on the GIT microbiota, which can help researchers draw more educated comparisons between studies.

Based on the microbial composition data obtained in this study and the effects reported in our previous publication [13], we speculate that the high-flavonoid corn diet modulated the microbiota in the small intestines by decreasing the population of pathogenic *C. perfringens* and increasing beneficial bacteria, which may have prevented epithelial adhesion and multiplication of toxin-producing *C. perfringens*. However, further investigation is needed to elucidate the effects of PennHFD1 on *E. maxima* infection, the immune response, as well as on the dietary conditions that alter the host's physiology to better understand how functional feedstuffs improve health in experimental infection models.

In conclusion, the co-infection of *E. maxima* and *C. perfringens* and the dietary high-flavonoid corn (PennHFD1) were associated with alterations in the GIT microbial diversity and composition of broiler chickens. *C. perfringens* was less differentially abundant in jejunal mucosal samples, and this was associated with a lower incidence of NE lesions. This study also supported previous knowledge that collecting different gastrointestinal tract segments and different sample types provides a more comprehensive understanding of the microbiota changes among treatments.

## Supporting information

**S1 Table. Effect of the main effects (feed and infection) and interactions on alpha diversity indices in the jejunal (JLC and JM) and ileal samples (ILC and IM) collected from 21 day-old chickens coinfected with *E. maxima* and *C. perfringens*.** ASV = amplicon sequence variant; Feed = Feed A (commercial corn line-based diet) or Feed B (PennHFD1-based diet); Infection = Co-infection with *E. maxima* and *C. perfringens* or control (not infected). (DOCX)

**S2 Table.** Mean relative abundance (%) of taxonomic groups identified as relatively abundant at the genus and species level in jejunal (JLC and JM) and ileal (ILC and IM) samples collected from infected (co-infection with *E. maxima* and *C. perfringens*) and control (non-infected) chickens fed a commercial corn-based diet (A) or a high-flavonoid corn-based diet (B). CTL A (Non-infected chickens fed a commercial corn-based diet); CTL B (Non-infected chickens fed a PennHFD1-based diet); INF A (Chickens co-infected with *E. maxima* and *C. perfringens* fed a commercial corn-based diet); INF B (Chickens co-infected with *E. maxima* and *C. perfringens* fed a PennHFD1-based diet). * Statistical difference (ANOVA, $P \leq 0.05$). (DOCX)

## Acknowledgments

The authors wish to thank Ms. Lori Schreier and Ms. Beverly Russell for laboratory support. Mention of trade name, proprietary product, or specific equipment does not constitute guarantee or warranty by USDA and does not imply its approval to the exclusion of other suitable products.

## Author Contributions

**Conceptualization:** Vinicius Buiatte, Monika Proszkowiec-Weglarz, Surinder Chopra, Mark Jenkins, Alberto Gino Lorenzoni.

**Data curation:** Vinicius Buiatte, Monika Proszkowiec-Weglarz, Katarzyna Miska.

**Formal analysis:** Vinicius Buiatte, Monika Proszkowiec-Weglarz, Katarzyna Miska.

**Funding acquisition:** Monika Proszkowiec-Weglarz, Katarzyna Miska, Surinder Chopra, Alberto Gino Lorenzoni.

**Investigation:** Vinicius Buiatte, Monika Proszkowiec-Weglarz, Katarzyna Miska, Alberto Gino Lorenzoni.

**Methodology:** Vinicius Buiatte, Monika Proszkowiec-Weglarz, Katarzyna Miska, Dorian Dominguez, Mahmoud Mahmoud, Tyler Lesko, Bryan P. Panek, Surinder Chopra, Mark Jenkins, Alberto Gino Lorenzoni.

**Project administration:** Vinicius Buiatte, Alberto Gino Lorenzoni.

**Resources:** Monika Proszkowiec-Weglarz, Katarzyna Miska, Surinder Chopra, Alberto Gino Lorenzoni.

**Supervision:** Monika Proszkowiec-Weglarz, Surinder Chopra, Alberto Gino Lorenzoni.

**Validation:** Vinicius Buiatte.

**Visualization:** Vinicius Buiatte, Monika Proszkowiec-Weglarz.

**Writing – original draft:** Vinicius Buiatte, Monika Proszkowiec-Weglarz.

**Writing – review & editing:** Vinicius Buiatte, Monika Proszkowiec-Weglarz, Katarzyna Miska, Dorian Dominguez, Mahmoud Mahmoud, Tyler Lesko, Bryan P. Panek, Surinder Chopra, Mark Jenkins, Alberto Gino Lorenzoni.

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
