## [Decision Letter · Decision Letter 0]

18 Apr 2024

PONE-D-24-09821The effects of a high-flavonoid corn cultivar on the gastrointestinal tract microbiota in chickens undergoing necrotic enteritisPLOS ONE

Dear Dr. Lorenzoni,

Thank you for submitting your manuscript to PLOS ONE. After careful consideration, we feel that it has merit but does not fully meet PLOS ONE’s publication criteria as it currently stands. Therefore, we invite you to submit a revised version of the manuscript that addresses the points raised during the review process.

Both reviewers remarked on the small sample sizes used here which led to both questioning the relevance and depth of the Discussion. I would also expect that the authors direct their revisions specifically to the following comments by the reviewers: Please address the potential direct effects of the fed additive on the coccidial infection and the outcome of the experiments.  Further, both reviewers suggest a more detailed Discussion and an explanation concerning correlation of the microbiota data found here versus causation!

We look forward to receiving your revised manuscript.

Kind regards,

Michael H. Kogut, Ph.D.

Academic Editor

PLOS ONE

Journal Requirements:

“This research was partially supported by a Hatch project (PEN04613) to SC and a Seed Grant Award from the College of Agricultural Sciences to GL and SC, and the in-house USDA-ARS CRIS project number 8042-31000-108-00D (MPW and KM).”

3. PLOS requires an ORCID iD for the corresponding author in Editorial Manager on papers submitted after December 6th, 2016. Please ensure that you have an ORCID iD and that it is validated in Editorial Manager. To do this, go to ‘Update my Information’ (in the upper left-hand corner of the main menu), and click on the Fetch/Validate link next to the ORCID field. This will take you to the ORCID site and allow you to create a new iD or authenticate a pre-existing iD in Editorial Manager. Please see the following video for instructions on linking an ORCID iD to your Editorial Manager account: https://www.youtube.com/watch?v=_xcclfuvtxQ.

Additional Editor Comments (if provided):

Reviewer #1: The results are interesting but there is a need to expand the discussion besides what they were expecting regarding the microbiota modulation. I see an important limitation in the NE challenge model regarding the interpretation of "mechanism of action". Coccidia is also being affected by the flavonoids, and it is at least the 50% of the changes that they are introducing there!

The study is a based in a challenge model that uses infection with coccidia as a predisposing factor for the development of C. perfringens necrotic enteritis. Although this is a very common NE challenge model used in several studies, and also imposed to the industry as almost a gold standar, the inclusion of feed additives or diet to compare treatments against a non-challenged control must be carefully interpreted. Many of these additives (or diet components) can alter the infection of coccidia (either in the level as well as the cycle (i.e. shortening or extending)) by direct interaction with the parasite or the physiology of the host. Particularly flavonoids are able to inhibit the invasion and replication of different species of coccidian and also alter epithelial physiology and immune responses of the gut. As C. perfringens challenge is synchronized at a fix time after coccidian delivery, if cycling is lightly altered, it can influence how C. perfringens induce lesions. I would suggest to include this discussion in the paper, limiting the expected outcomes of the results obtained.

In lines 481-483 the authors state that the results "indicate" that microbiota modulation is an important mechanism of action of high-flavonoid corn in the observed reduction of necrotic enteritis. However, although the results show a modulation of the microbiota, they are not strong enough, due to the experimental design and the co-infection challenge model used, to "indicate" that it is a mechanism of action. The results do show that there is a change in the microbiota and that these changes are associated with a decrease in the number (?) of animals with lesions compatible with necrotic enteritis (I understand that there would be no differences in the degree of lesion, the authors refer to a previous work that is supposed to continue in this one) but there is no direct relationship. These results could suggest a potential role of the microbiota in the observed effects. However, if flavonoids are altering for example intestinal transit, water homeostasis, mucus secretion or antimicrobial peptides secretion, then a microbiota modulation would be also observed. Please consider the potential changes in animal physiology as part of the mechanisms of action of the high-flavonoid corn on the observed microbiota changes.

Reviewer #2:

1. The number of replicates is very small to make any logical conclusions.

2. provide the flavainoid content of the corn.

3. provide the basic nutritive value of the corn under study.

Reviewers' comments:

Reviewer's Responses to Questions

**Comments to the Author**

1. Is the manuscript technically sound, and do the data support the conclusions?

Reviewer #1: Partly

Reviewer #2: Yes

2. Has the statistical analysis been performed appropriately and rigorously? 

Reviewer #1: Yes

Reviewer #2: Yes

3. Have the authors made all data underlying the findings in their manuscript fully available?

Reviewer #1: Yes

Reviewer #2: Yes

4. Is the manuscript presented in an intelligible fashion and written in standard English?

Reviewer #1: Yes

Reviewer #2: Yes

5. Review Comments to the Author

Reviewer #1: The manuscript “The effects of a high-flavonoid corn cultivar on the gastrointestinal tract microbiota in chickens undergoing necrotic enteritis” show results that are interesting and useful to understand the evolution of necrotic enteritis and how this type of corn can induce changes in the animal. The study is a based in a challenge model that uses infection with coccidia as a predisposing factor for the development of C. perfringens necrotic enteritis. Although this is a very common NE challenge model used in several studies, and also imposed to the industry as almost a gold standar, the inclusion of feed additives or diet to compare treatments against a non-challenged control must be carefully interpreted. Many of these additives (or diet components) can alter the infection of coccidia (either in the level as well as the cycle (i.e. shortening or extending)) by direct interaction with the parasite or the physiology of the host. Particularly flavonoids are able to inhibit the invasion and replication of different species of coccidian and also alter epithelial physiology and immune responses of the gut. As C. perfringens challenge is synchronized at a fix time after coccidian delivery, if cycling is lightly altered, it can influence how C. perfringens induce lesions. I would suggest to include this discussion in the paper, limiting the expected outcomes of the results obtained.

In lines 481-483 the authors state that the results "indicate" that microbiota modulation is an important mechanism of action of high-flavonoid corn in the observed reduction of necrotic enteritis. However, although the results show a modulation of the microbiota, they are not strong enough, due to the experimental design and the co-infection challenge model used, to "indicate" that it is a mechanism of action. The results do show that there is a change in the microbiota and that these changes are associated with a decrease in the number (?) of animals with lesions compatible with necrotic enteritis (I understand that there would be no differences in the degree of lesion, the authors refer to a previous work that is supposed to continue in this one) but there is no direct relationship. These results could suggest a potential role of the microbiota in the observed effects. However, if flavonoids are altering for example intestinal transit, water homeostasis, mucus secretion or antimicrobial peptides secretion, then a microbiota modulation would be also observed. Please consider the potential changes in animal physiology as part of the mechanisms of action of the high-flavonoid corn on the observed microbiota changes.

Reviewer #2: !. The number of replicates is very small to make any logical conclusions. But it is what it is.

2. provide the flainoid content of the corn.

3. provide the basic nutritive value of the corn under study.

6. PLOS authors have the option to publish the peer review history of their article (what does this mean?). If published, this will include your full peer review and any attached files.

Reviewer #1: **Yes: **mariano fernandez miyakawa

Reviewer #2: No

---

## [Author Response · Author response to Decision Letter 0]

1 Jul 2024

A formatted version of this letter is included with all the other files. We could not copy paste tables in this field.

Manuscript: PONE-D-24-09821

Dear Dr. Kogut, PLOS ONE Academic Editor,

We are pleased to submit the revised draft of our manuscript “The effects of a high-flavonoid corn cultivar on the gastrointestinal tract microbiota in chickens undergoing necrotic enteritis” for consideration of publication. We are thankful for the opportunity to receive constructive feedback that helped improve our manuscript. We believe that we were able to address most of your concerns. 

Please see below a point-by-point response to the reviewers’ comments. Our answers in this letter are highlighted in blue color, and the changes made to the manuscript are highlighted in yellow.

Sincerely,

Alberto Gino Lorenzoni, DVM, MS, PhD.

Associate Professor of Poultry Science and Avian Health

Department of Animal Science

The Pennsylvania State University

Corresponding Author

Reviewer’s comments to the authors:

Reviewer #1

The results are interesting but there is a need to expand the discussion besides what they were expecting regarding the microbiota modulation. I see an important limitation in the NE challenge model regarding the interpretation of "mechanism of action". Coccidia is also being affected by the flavonoids, and it is at least the 50% of the changes that they are introducing there!

The study is a based in a challenge model that uses infection with coccidia as a predisposing factor for the development of C. perfringens necrotic enteritis. Although this is a very common NE challenge model used in several studies, and also imposed to the industry as almost a gold standar, the inclusion of feed additives or diet to compare treatments against a non-challenged control must be carefully interpreted. Many of these additives (or diet components) can alter the infection of coccidia (either in the level as well as the cycle (i.e. shortening or extending) by direct interaction with the parasite or the physiology of the host. Particularly flavonoids are able to inhibit the invasion and replication of different species of coccidian and also alter epithelial physiology and immune responses of the gut. As C. perfringens challenge is synchronized at a fix time after coccidian delivery, if cycling is lightly altered, it can influence how C. perfringens induce lesions. I would suggest to include this discussion in the paper, limiting the expected outcomes of the results obtained.

Thanks for taking the time to thoroughly review our manuscript and give us feedback. 

We agree with your comments. We expanded the discussion with supporting literature that considers the other factors that interplay in the effects observed in our study. (Page 21, lines 445-457).

In lines 481-483 the authors state that the results "indicate" that microbiota modulation is an important mechanism of action of high-flavonoid corn in the observed reduction of necrotic enteritis. However, although the results show a modulation of the microbiota, they are not strong enough, due to the experimental design and the co-infection challenge model used, to "indicate" that it is a mechanism of action. The results do show that there is a change in the microbiota and that these changes are associated with a decrease in the number (?) of animals with lesions compatible with necrotic enteritis (I understand that there would be no differences in the degree of lesion, the authors refer to a previous work that is supposed to continue in this one) but there is no direct relationship. These results could suggest a potential role of the microbiota in the observed effects. However, if flavonoids are altering for example intestinal transit, water homeostasis, mucus secretion or antimicrobial peptides secretion, then a microbiota modulation would be also observed. Please consider the potential changes in animal physiology as part of the mechanisms of action of the high-flavonoid corn on the observed microbiota changes.

We agree with your comment. We made changes to the text (Page 23, Lines 492-495; and Lines 497-502) and added supporting information (Page 21, lines 444-457) to explore the other factors that were not evaluated in this study.

Reviewer #2:

1. The number of replicates is very small to make any logical conclusions.

We refined our conclusion to account for pitfalls related to experimental design (Page 23, Lines 496-502).

2. provide the flavainoid content of the corn.

We added a sentence reporting the flavonoid content of both corn lines used in the study, and referenced previously reported data (Page 6, Lines 147-148).

Reference: 

Buiatte V, Dominguez D, Lesko T, Jenkins M, Chopra S, Lorenzoni AG. Inclusion of high-flavonoid corn in the diet of broiler chickens as a potential approach for the control of necrotic enteritis. Poult Sci. 2022;101. doi:10.1016/j.psj.2022.101796

3. provide the basic nutritive value of the corn under study.

The complete wet chemistry analyses of the corn lines have been previously reported (Figure below). We added a sentence in the materials and methods to address your concern. (Page 6, Lines 148-149).

Reference: 

Buiatte V, Dominguez D, Lesko T, Jenkins M, Chopra S, Lorenzoni AG. Inclusion of high-flavonoid corn in the diet of broiler chickens as a potential approach for the control of necrotic enteritis. Poult Sci. 2022;101. doi:10.1016/j.psj.2022.101796

Source: Buiatte et al. (2022), Poultry Science.

---

## [Editor Report · Decision Letter 1]

3 Jul 2024

The effects of a high-flavonoid corn cultivar on the gastrointestinal tract microbiota in chickens undergoing necrotic enteritis

PONE-D-24-09821R1

Dear Dr. Lorenzoni,

We’re pleased to inform you that your manuscript has been judged scientifically suitable for publication and will be formally accepted for publication once it meets all outstanding technical requirements.

Kind regards,

Michael H. Kogut, Ph.D.

Academic Editor

PLOS ONE
---

## [Editor Report · Acceptance letter]

9 Jul 2024

PONE-D-24-09821R1 

PLOS ONE

Dear Dr. Lorenzoni, 

I'm pleased to inform you that your manuscript has been deemed suitable for publication in PLOS ONE. Congratulations! Your manuscript is now being handed over to our production team.

Kind regards, 

on behalf of

Dr. Michael H. Kogut 

Academic Editor

PLOS ONE